

# OceanPrediction Decade Collaborative Center: Connecting the world around ocean forecasting

Enrique Alvarez Fanjul and Pierre Bahurel

Mercator Ocean International, Toulouse, France

*Correspondence to*: Enrique Alvarez Fanjul (ealvarez@mercator-ocean.fr)

**Abstract.** Operational Ocean Forecasting Systems (OOFS) have proven immensely valuable today. Numerous successful and inspiring services are currently operating in various world regions, contributing to cutting-edge applications within the marine
community. This success lays a strong foundation for building a global community around ocean forecasting. However, developing and enhancing new forecasting systems remains challenging due to the absence of best practices, standards, and community-endorsed architectures. The OceanPrediction Decade Collaborative Center (DCC) and its associated Decade actions aim to address these challenges by leveraging the UN Decade of Ocean Science and the concept of digital twinning. This paper introduces the OceanPrediction DCC and outlines the forward-looking strategies developed to achieve these
ambitious goals. The special issue introduced by this paper is part of this broader effort.

## 1 Introduction

The United Nations Decade of Ocean Science for Sustainable Development (2021-2030), also referred to as 'the Decade,' was proclaimed by the 72nd Session of the UN General Assembly on December 5, 2017. Coordinated by the IOC-UNESCO, the Decade seeks to promote large-scale, transformative change to shift from the 'ocean we have' to the 'ocean we want.' The
Decade supports the evolution of ocean data, information, and knowledge systems, driving them toward higher levels of readiness, accessibility, and interoperability. The scale of this effort must be exponentially greater than anything previously undertaken.

To guide the Decade's implementation, the IOC (Intergovernmental Oceanographic Commission) has developed an Implementation Plan (IOC-UNESCO, 2021), supported by contributions from member states, UN agencies, intergovernmental
organizations, non-governmental organizations, and relevant stakeholders. The OceanPrediction DCC is a cross-cutting structure within this plan that operates globally, fostering collaboration among Decade actions related to ocean prediction.

Mercator Ocean International has been entrusted by the IOC-UNESCO to coordinate the OceanPrediction DCC, with the mission: "to achieve a predicted ocean through a shared and coordinated global effort within the framework of the UN Ocean Decade." The Centre implements a community-driven agenda that allows the ocean prediction community to collaborate on
activities such as communication, outreach, training, cost-sharing, joint workshops, and the standardization of language and outputs. Additionally, it facilitates the co-design of an architecture necessary for developing a Global Ocean Prediction System.



The Centre acts as a global convener of multidisciplinary ocean prediction expertise, collaborating with intergovernmental programs (e.g., GOOS, ETOOFS, IODE, OBPS) to establish agreements on operational infrastructure, terminology, and standards needed to deliver unified services from multiple geographic and thematic nodes

## 2 OceanPrediction DCC objectives

The main objectives of the OceanPrediction DCC (https://www.unoceanprediction.org/en) are:

- To provide a collaborative backbone structure and a collective voice for the ocean prediction community, supporting the Decade's implementation, focusing on:
  - Creating a global, inclusive forum (spanning coastal to deep sea, nowcasting to climate, biology to physics, public to private, users to scientists) and other tools to facilitate dialogue and information exchange.
  - Implementing capacity development and ocean literacy initiatives.
  - Promoting OOFS as a crucial tool for the Blue Economy and ocean policy.
- To develop a global technical and organizational structure centered on:
  - Co-designing, in collaboration with Ocean Decade actions and other key stakeholders, a new scenario for ocean forecasting based on interoperability and an architecture that facilitates the "deliver as one" approach while leveraging digital twin technologies.
  - Identifying needs and coordinating the development of new tools, standards, and best practices for implementing this new scenario, with a focus on a science-to-service framework and promoting interoperability and integration.
  - Aligning Decade actions with the objectives of ocean forecasting and fostering collaboration between Decade initiatives and other relevant actors.
- To support the Decade Coordination Unit (head of the Decade) by collaborating with other Decade Collaborative Centers and Coordination Offices, ensuring alignment and monitoring of Decade actions to secure their long-term legacy.

## 3 OceanPrediction DCC in the UN "decade ecosystem"

OceanPrediction DCC will closely coordinate with the Data Sharing DCO (led by IODE) and the Observations DCO (led by GOOS) to establish a framework for developing ocean monitoring and forecasting services throughout the Decade. OceanPrediction DCC shall be responsible for promoting collaboration between Decade Programmes, and their relevant decade projects, as well as decade contributors when these fall under the scope of work, all done in coordination with the mentioned DCOs.





The Decade implementation plan links each DCC and DCO to specific Decade Programmes, named "primary attachments". In the case of OceanPrediction DCC, these are:

- ForeSea – The Ocean Prediction Capacity of the Future
- Ocean Practices
- Digital Twins of the Ocean (DITTO)
- Global Environment Monitoring System for the Ocean and Coasts (GEMS Ocean)
- Ocean Acidification Research for Sustainability
- NASA Sea Level Change Science Team
- France's Priority Research Program "Ocean of Solutions"

The collaboration with these programmes will be particularly intensive, but additional collaborations with other programmes will be established, as "secondary attachments".

## 4 OceanPrediction DCC collaborative structure

To achieve its objectives, OceanPrediction DCC will establish two global collaboration structures:

- A central structure, comprising the Ocean Forecasting Global Co-design Team (OFCT) and a central office, which will liaise with various UN, EU, and national bodies. The OFCT focuses on co-design alignment and consists of experts covering different aspects of the ocean forecasting value chain (Alvarez Fanjul et al., 2022).
- A decentralized regional structure, consisting of Regional Teams that focus on community development and capacity-building efforts.

Having different Teams for technical aspects and community building will allow efficient management: a smaller specialists team able to deliver technical results on time and a larger geographically based structure, able to integrate the community and catalyze the governance and organizational component.

### 4.1 The Regional teams

The OceanPrediction DCC Regional teams have the following objectives:

- Act as regional nodes of OceanPrediction DCC
- Contribute to the coordination and cooperation with ocean forecasting-related Decade actions in the region.
- Identify gaps and ways forward in the regional landscape of ocean forecasting.
- Support OceanPrediction DCC in the design and organization of regional events for capacity building, ocean literacy, and other purposes, such as courses, workshops, hackathons, etc.
- Advocate for regional implementation of Best Practices, Standards, and Tools derived from OceanPrediction activity.





- Collaborate with the other OceanPrediction DCC Regional Teams to support global actions
- Support OceanPrediction DCC in obtaining information for the three Atlases (services, institutions, interested persons, experts) and any other relevant data.
- Promote the use of OOFS in each region for decision-making purposes, including sustainable blue economy, technical, policy, and legal aspects.

The Regional Team distribution is based both on UNEP (United Nations Environment Programme) regional seas and in GOOS Regional Alliances (GRAs), clustering some regions. The concept of the Regional Teams was announced at the OceanPrediction DCC kick-off meeting, an event that demonstrated the appetite for this initiative, with 1800 registered
participants from all continents. At this moment we are building these teams, and several leaders are volunteering worldwide to chair each region:

- Region 1: West Pacific and Marginal Seas of South and East Asia. Chair: Swadhin Behera (JAMSTEC-Japan)
- Region 2: Indian seas, covering South Asian Seas and ROPME Sea Area. Chair: Sudheer Joseph (INCOIS-India)
- Region 3: African seas. Chair: Kouadio Affian (Ivory Coast - Chair of IOC Africa). For this region, we have decided
to have several co-chairs and a subregional division to address the differences in technical development.
- Region 4: Mediterranean and Black Sea. Chair: Emanuela Clementi (MONGOOS/CMCC - Italy)
- Region 5: North-East Atlantic. Chairs: Ghada al Serafy and Loreta Cornacchia (EuroGOOS coastal WG, Deltares)
- Region 6: South and Central America: Chairs:
- Region 7: North America: Chairs: Patrick Hogan (NOAA), and Fraser Davidson (DFO).
- Region 8: Arctic: Chair: Heather Reagan (NERSC-Norway)
- Region 9: Antarctic: Chair: Stuart Corney (UTAS - Australia)

**4.2 The Ocean Forecasting Co-Design Team**

Ocean Forecasting Systems (OFS) have proven invaluable for understanding the ocean and providing critical information for decision-making. However, challenges remain in areas like standardization, interoperability, and integration. Building an OFS
from scratch, without guidance, is a daunting task, often resulting in isolated systems with limited integration into a larger framework.

This situation hampers the proliferation of forecasting services, especially in technologically less advanced countries, and hinders the growth of the ocean forecasting community and collective knowledge. The Ocean Forecasting Co-Design Team (https://www.unoceanprediction.org/en/about/technical) is an international group of experts working under OceanPrediction
DCC coordination, collaborating to overcome these limitations by developing a new ocean forecasting architecture based on the digital twin concept. This team is composed of worldwide specialists whose collective expertise covers the whole value chain.





As an initial step, the team has assembled the current special issue and evaluated the status of operational ocean forecasting
systems from both user and expert perspectives (Ciliberti et al., 2023). The team's primary objective is to design a unified
ocean forecasting architecture that leverages the concept of digital twinning. This architecture aims to facilitate simpler,
modular, and more robust system development in the future. A key aspect of this development will be the establishment of
well-defined building blocks, which will take the form of standards, tools, and best practices. While this new framework will
benefit all forecasting services, it will be especially impactful for organizations that are just beginning their activities.

The Ocean Forecasting Co-Design Team's role is to identify this architecture and define the essential building blocks needed
for its expansion. This effort will support the various Decade Programmes by providing clear development targets. However,
the team's role is not to "code" these components directly, but rather to inspire and guide the implementation of these targets
by Decade Programmes.

## 5 Next Steps

The OFCT will continue its activities, and, in the future, it is planned to address the identification of gaps in ocean forecasting
and the priorities for further development. The results of these works will be published in subsequent special issues. These
efforts form part of a wider strategy to promote ocean forecasting worldwide, which is summarized in the virtuous loop shown
in **Figure 1**.

The Ocean Prediction DCC's community, organized around the regional teams and integrating the Decade Programmes related
to Ocean Forecasting, will be at the center of all the developments. This community will be articulated through the
OceanPrediction DCC web page (https://www.unoceanprediction.org/en) and, more specifically, around a Forum, where the
community will share experiences and address doubts, and an Atlas, that will serve to identify who is who.



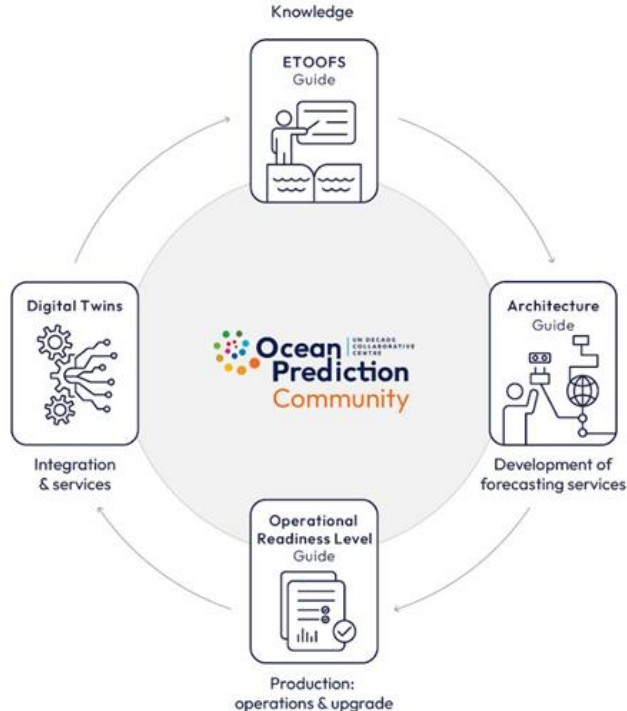

**Figure 1: OceanPrediction DCC's Virtuous loop towards the promotion of Ocean Forecasting**

The virtuous loop starts with the knowledge required to understand ocean forecasting techniques and their degree of
development and implementation. The publications here presented, and the future gap analysis mentioned above are part of
this effort, which is centralized around the ETOOFS guide. This GOOS publication (Alvarez-Fanjul et al., 2022) compiles the
basic knowledge related to the different aspects of Ocean Forecasting. Now the Guide has been transformed into a wiki site
under the OceanPrediction DCC website. This will permit the update of content by the addition of community contributions.

This common knowledge will lead to the construction of new operational services and the improvement of existing ones. To
facilitate this task, the OFCT will soon deliver the so-called "Architecture Guide". This document will describe all the
components, and "internal wiring" required to implement a robust forecasting service. The architecture is based on "building
blocks", which will take the form of Data Standards and Tools.

Once a system is implemented, it is required to operate it properly. To facilitate this task, the OFCT has developed the
Operational Readiness Level (ORL), to be published soon. This is a new tool to promote the adoption and implementation of
best practices in ocean forecasting. Thanks to its application, system developers will be able to assess the operational status of
an ocean forecasting system. Improving the ORL qualification of a service is a means to implement best practices and standards
in ocean forecasting, improving the system.

The ORL comprises three independent digits designed to certify the operational status of an ocean forecasting system. Each
digit ranges from 0 (minimum) to 5 (maximum), with decimal numbers being allowed. These digits correspond to distinct





aspects related to operationality: the First Digit reflects the reliability of the service, the Second monitors the level of validation for the service, and the Third assesses the various degrees of product dissemination achievable by the system.

In the last conceptual step of the virtuous loop, the data will be integrated into interoperable frameworks, such as Digital Twins of the Ocean. This will allow a richer exploitation of the data, extracting more information useful for science and decision-making. The knowledge generated in this way will be incorporated into our common, closing the loop.

We intend that this compilation becomes a relevant part of the shared knowledge that forms part of this loop, describing where ocean forecasting stands today. By examining current methods and new developments, we highlight how important ocean forecasting is for keeping our marine environment healthy and productive for future generations.

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

**Competing interests**

The contact author has declared that none of the authors has any competing interests.

**Data and/or code availability**

This can also be included at a later stage, so no problem to define it for the first submission.



**Authors contribution**

This can also be included at a later stage, so no problem to define it for the first submission.

**Acknowledgements**

This can also be included at a later stage, so no problem to define it for the first submission.