# Peer review of "OceanPrediction Decade Collaborative Center: Connecting the world"

_State of the Planet, 2024_

## Referee Comment (RC1)

[referee-annotated manuscript omitted]

---

## Author Response (AR1)

**Review #1**

The manuscript "OceanPrediction Decade Collaborative Center: Connecting the World Around Ocean Forecasting" is well-written and contains vital information about international efforts to co-design state-of-the-art operational ocean forecasting systems. The manuscript describes the conceptual framework of OceanPredict DCC and its goals and aims for the near future. I am sure this will be an essential contribution to future operational systems.

Minor comments.

In many places, capital letters are used, which may be addressed. I have added minor edits and comments at relevant points as annotations in the attached PDF file.

**Answer #1**

Dear Reviewer,

Thank you very much for your positive review and your (minor) comments on the document. I have looked into them and will introduce all the changes you suggest.

Best Regards,

Enrique

**Review #2**

This article by Fanjul and Bahurel gives a useful overview of the DCC. I only have a few small comments on passages that I find unclear.

line 11: "developing and enhancing new" - if "new" I do not see a need for enhancing. If you do something new you can add whatever you feel is needed. Maybe you mean "developing new and enhancing existing"?

line 13: write "UN Decade" full name please

line 28: "to achieve a predicted ocean through" - does this include all time scales? in particular I mean climate predictions. If not climate - please precise here that you do not address efforts such as IPCC.

Line 46: "facilitates the "deliver as one" approach" - what is that approach? can you add a reference to clarify? sounds like people in your community know that phrase.

line 49: "this new scenario" - what is meant by "Scenario"?

Line 63: Would it make sense to enrich the list of decade programm names (which are pretty meaningless as such) with mission statements?

Line 78: "Regional Teams" you augment to "Regional Teams ( see below) ..."

Line 93: "three Atlases": three Atlases? what is meant with Atlases? or you mean the FOUR words in the bracket that follows?

Line 108: something missing after "Chair:"

Line 114: "challenges remain in areas like standardization, interoperability, and integration." If you include climate predictions in your efforts you may want to mention that in the IPCC model world has very sophisticated and proven approaches for standardization, interoperability and integration and that these approaches may be used for guiding the way forward in other forecast sectors? If climate predictions are not covered by your efforts you may still want to refer to the approaches developed by IPCC / UNFCCC as "best practices"? ("best" - because the approaches in the IPCC model world have been developed from a global community)

Line 125: could you add a reference for the "digital twinning" otherwise this is only a buzzword and kind of meaningless

Line 134: "gaps" - similar comment as for "digital twinning" (Line 125)

Line 144: A loop does not "start" this is by definition for a loop - maybe say "beginning the loop with…."

Line 145: "The publications here presented" - you mean with "here presented" in this special issue? write so - if that is what you mean - and clarify if this is not what you mean.

Line 146: "...around the ETOOFS guide. This GOOS publication (Alvarez-Fanjul et al., 2022)" - Is this ETOOFS guide the GOOS publication?

This is confusing - make use of standard referencing as it is best practices in all journals - or introduce the term "GOOS publication" as part of a reference. e.g. ...around the ETOOFS guide (Alvarez-Fanjul et al., 2022, from now on called "GOOS publication")...

Line 149: "This common knowledge will lead to the construction..." - what make you think that a wiki will lead to create new services? someone has to be a) convinced approaches are better and b) time, money, spending effort to implement them. Or is the "Architecture guide" a summary of the wiki that is communicated to communities? if so - write that connection explicitly; if not - how will you propmt the community members to use the wiki?

Line 154: "Operational Readiness Level (ORL), to be published soon" - maybe say where it will be published? e.g. as part of this collection?

**Answer #2**

Dear Reviewer,

Thank you very much for your positive review and your (minor) comments on the document. I have looked into them and will introduce all the changes you suggest.

Best Regards,

Enrique